# Tuning instability in suspended monolayer 2D materials

Yuan Hou[1,2,4], Jingzhuo Zhou [2,4], Zezhou He[3,4], Juzheng Chen[2], Mengya Zhu[2], HengAn Wu [3] & Yang Lu [1] ✉

Monolayer two-dimensional (2D) materials possess excellent in-plane mechanical strength yet extremely low bending stiffness, making them particularly susceptible to instability, which is anticipated to have a substantial impact on their physical functionalities such as 2D-based Micro/Nanoelectromechanical systems (M/NEMS), nanochannels, and proton transport membrane. In this work, we achieve quantitatively tuning instability in suspended 2D materials including monolayer graphene and $MoS_2$ by employing a push-to-shear strategy. We comprehensively examine the dynamic wrinkling-splitting-smoothing process and find that monolayer 2D materials experience stepwise instabilities along with different recovery processes. These stepwise instabilities are governed by the materials' geometry, pretension, and the elastic nonlinearity. We attribute the different instability and recovery paths to the local stress redistribution in monolayer 2D materials. The tunable instability behavior of suspended monolayer 2D materials not only allows measuring their bending stiffness but also opens up new opportunities for programming the nanoscale instability pattern and even physical properties of atomically thin films.

Two-dimensional (2D) materials have attracted considerable interest because of many striking mechanical and physical properties that originated from the unique planar atomic structure and bonding nature[1,2]. However, suspended 2D materials, especially for their monolayer, can hardly withstand compressions due to the ultralow bending resistance[3]. That means the 2D planar structure will undergo out-of-plane deformations, such as rippling, buckling, wrinkling, or even creasing, when triggered by geometrical constraints[4], which have significant influences on the mechanical[5,6], electrical[7], and thermal[8] properties of 2D materials. Their mechanical stability, therefore, becomes especially important in those device applications including suspended 2D-based Micro/Nanoelectromechanical systems (M/NEMS)[9], resonator/oscillator[10], kirigami/origami[11], electrodes[12] and nanochannels[13–15]. Recently, the wrinkles induced by instability in monolayer graphene were demonstrated to accelerate proton transport[16], which has renewed demands in tuning of instability morphology of suspended 2D materials. However, there is a lack of in situ characterization and quantitatively mechanical measurement of instability and recovery process in suspended monolayer 2D materials since previous experiments normally relied on substrate-supported instability behavior in which the interface masks the subtle intrinsic mechanics of 2D materials[4].

Experimentally, applying controllable out-of-plane deformation for creating and tuning the instability in suspended 2D materials is rather challenging. Previous work firstly achieved controllable buckling of suspended graphene layers by controlling boundary conditions and utilizing graphene's negative thermal expansion coefficient[17]. They further observed both compression- and shearing-induced static periodic wrinkles and wrinklons (which referring to the initial wrinkles could split into period-halving wrinkles[18]) in the buckled graphene,

[1]Department of Mechanical Engineering, The University of Hong Kong, Pokfulam 999077 Hong Kong SAR, China. [2]Department of Mechanical Engineering, City University of Hong Kong, Kowloon 999077 Hong Kong SAR, China. [3]CAS Key Laboratory of Mechanical Behavior and Design of Materials, Department of Modern Mechanics, University of Science and Technology of China, Hefei 230027, China. [4]These authors contributed equally: Yuan Hou, Jingzhuo Zhou, Zezhou He. ✉e-mail: ylu1@hku.hk

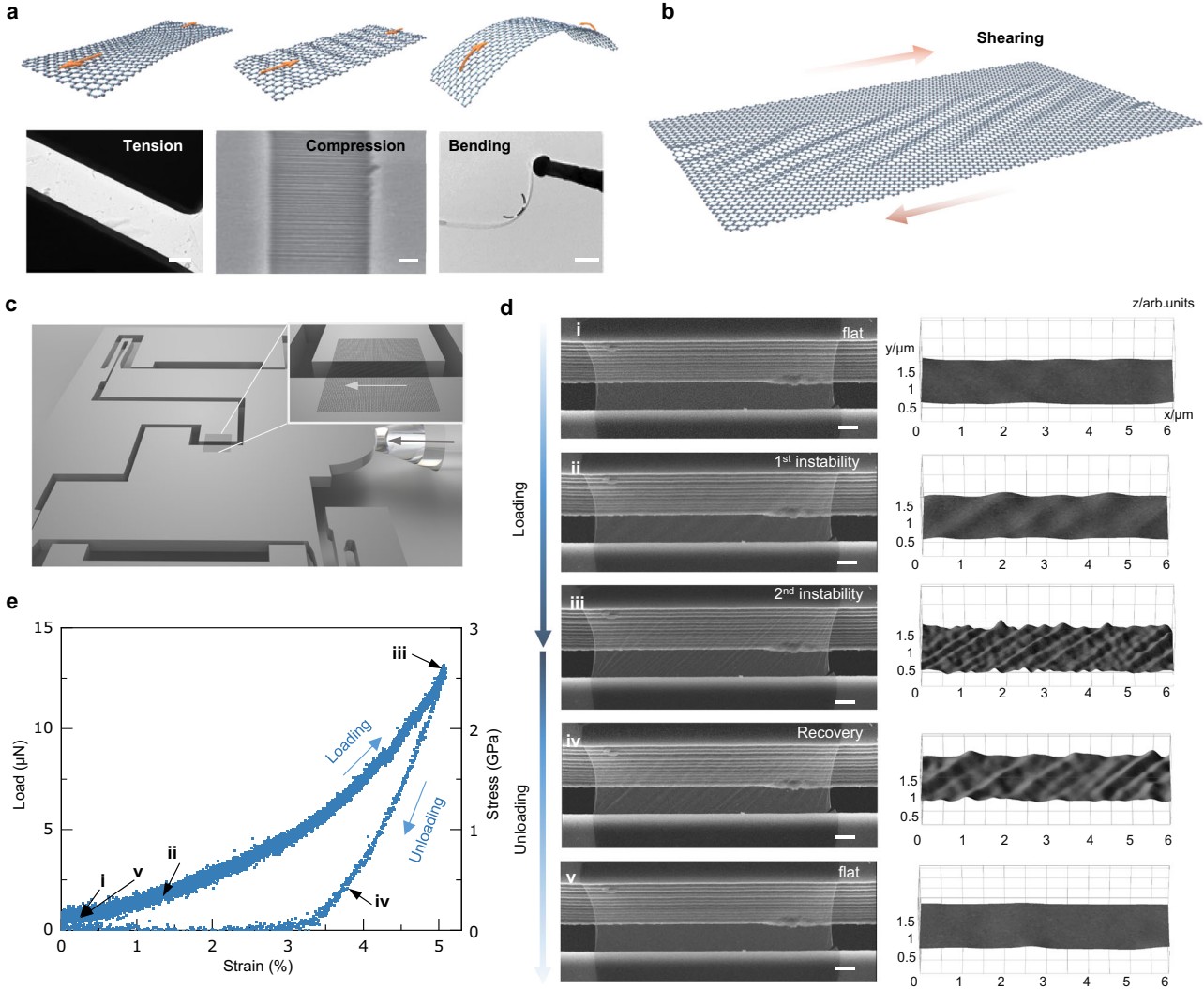

**Fig. 1 | Experimental setup and tuning instability of monolayer graphene.** The schematics and corresponding electron microscope images illustrate four fundamental mechanical experiments for 2D materials: **a** tension, compression, bending, and **b** shearing. (Reproduced with permissions from ref. 17, Springer Nature Limited; ref. 25, American Physical Society; ref. 26, CC BY 4.0). **c** The experimental setup of in-plane shearing of monolayer 2D materials incorporates a mechanically push-to-shear design. **d** In situ images of shear-induced reversible instability of monolayer graphene and the corresponding 3D morphologies. **e** Shear force (stress)–strain curve during the loading-unloading process, where the arrows indicate the emergences and disappears of the instabilities. Scale bars: **a** 1 μm, 500 nm, and 200 nm, and **d** 1 μm.

which is induced by spontaneously and/or thermally generated strains[17]. Similar phenomena were also discovered in other thin films[18–21]. However, the instability and recovery process of suspended monolayer 2D materials cannot yet be well quantified due to the absence of mechanical measurements. Besides, the transition between wrinkles and wrinklons, which plays a vital role in tuning instability behaviors of thin films[22–24], remains elusive in suspended monolayer 2D materials. To achieve controllable instability in suspended 2D materials, several critical questions need to be answered. What are the critical instability conditions for suspended monolayer 2D materials? How can we regulate their instability morphology? After unloading, does the recovery process backtrack the forward process? Quantitative experimental measurement of 2D materials instability is therefore urgently needed.

In this work, we implement in situ shear loading-unloading of suspended monolayer 2D materials, including graphene and molybdenum disulfide (MoS₂), by employing a mechanical push-to-shear (PTS) strategy, which allows precise tuning of the emergence and evolution of instability patterns. Compared to the realized micro/nano-scale mechanical experiments for 2D materials, e.g., tension,

compression, and bending experiments (Fig. 1a)[17,25,26], the in-plane shearing can precisely control the instability behavior of thin film. In a broad sense, shearing properties of 2D materials (Fig. 1b), become increasingly important in 2D material-based systems (nanocomposites[27,28], twisted moiré materials[29,30], and resonators[31], etc.) since in-plane shear deformation significantly affects the instability behaviors of 2D materials[32] (which have always been underappreciated due to the lack of nanoscale shear measurements). In our measurement, we observed the rise of primary and secondary instabilities under increasing shear loadings. The loadings can be quantified by a series of critical stresses governed by pretension, geometry, and modulus of materials. We further reveal a scaling law $D \sim f(E, \varepsilon_{pre}, \gamma)\lambda^4$ to quantitatively relate the bending stiffness, Young's modulus, strains, and wrinkling wavelength, by which we demonstrate a protocol to measure the bending stiffness of monolayer 2D materials. The dynamic shearing process indicates that the instability involves a step-by-step wrinkle-splitting behavior that is dominated by wavelength decreasing. While the recovery process involves a steady amplitude-decreasing-dominated smoothing rather than wrinkle merging. Such different instability and recovery trajectories can be

explained by the local compressive stress redistribution of monolayer 2D materials. Tuning the instability in monolayer 2D materials could contribute to programming their morphology and further expand the physical functionalities of suspended 2D-based devices.

## Results

### Regulation of instability morphology in suspended monolayer 2D materials

To conduct in situ shear experiments for monolayer 2D materials, we developed a probe- and capillary-assisted transfer method to fabricate the suspended specimen onto a PTS micro-mechanical device (MMD) (Supplementary Figs. 1-3). This mechanical design allows us to precisely control the shearing loading-unloading process of 2D materials through the electrostatic actuator. As shown in Fig. 1c, the monolayer graphene was placed on the gap between adjacent silicon cantilevers. The upper silicon cantilever (Fig. 1c) forces the graphene film laterally translation after the diamond-fabricated indenter contacts the top of the MMD, while the bottom cantilever remains stationary, inducing the 2D materials on the gap to produce in-plane shear deformation. To more clearly capture the contrast change induced by out-of-plane deformation, we set an additional tilt angle for the PTS MMD so that at this viewing angle, a portion of the MMD's side wall is exposed (Detailed procedures are given in Method). Combined with scanning electron microscopy (SEM), the morphology of 2D materials and the force-displacement curves can be recorded simultaneously.

Figure 1d shows a typical shearing-induced wrinkling behavior of monolayer graphene (the layer number was confirmed by Raman spectra; see Supplementary Fig. 2). The morphology of monolayer graphene displayed a reversible smoothing-wrinkling-smoothing phenomenon throughout the loading-unloading process: (i-ii) Wrinkling appeared (primary or $1^{st}$ instability) as indicated by the contrast change (qualitatively correlated with the out-of-plane deformation); (ii-iii) Once the shear strain increased over a certain threshold, the wavelength of the wrinkles dropped with a sequence of splitting, which is named as the secondary or $2^{nd}$ instability; (iii-v) The wrinkling regions recovered to a smooth state but without merging during the unloading stage. The detailed process is provided in Supplementary Movie 1. In addition, based on a simple analysis of in situ SEM images, the boundary slippage was negligible in our experiments (Supplementary Fig. 4).

Overall, the planar structure of monolayer graphene becomes unstable and forms periodic wrinkling structures upon shearing. This tunable instability process was reversible since the wrinkles could be smoothed thoroughly. However, the smoothing process showed different trajectories. We also performed shear experiments of monolayer $MoS_2$ (Supplementary Fig. 5, and Movie 2), which showed a similar reversible instability compared to graphene, while the wavelength evolution of $MoS_2$ differs from that of graphene. We measured the shear strain of graphene from the ratio of relative displacement between two sides and the width of the gap ($\gamma = \delta/W$). The shear stress could be calculated by indentation force dividing the cross area of film ($\tau = F/Lt$, $L$ represents the width of 2D materials and $t$ represents the thickness, typically assumed as 0.34 nm for graphene and 0.65 nm for $MoS_2$). The shear stress-strain curve of monolayer graphene is shown in Fig. 1e, where the in-plane shear modulus ($G$) of monolayer graphene was determined as ~70 GPa via the initial linear stage of the stress-strain curve; to date, such value is first measured by direct mechanical test[31]. Here, the hysteresis in the loading-unloading curves is due to the experimental setup, as explained in Supplementary Fig. 6. Besides, the low shear modulus measured in our experiment may be attributed to the fact that films are typically subjected to initial corrugations. Note that, Fig. 1e depicts successive stress increasing during the primary instability, but the stress during the secondary instability showed several sudden drops. Then in the unloading stage, the stress decreased continuously without breaks. We will later leverage the

stress drops to explain the differences in instability and recovery paths. Before that, we discuss the rich mechanics of the instability process.

### Primary instability and bending stiffness measurement

Figure 2a shows the typical snapshots of monolayer graphene at a series of shear strains before wrinkling splitting (the detailed process is shown in Supplementary Movie 3). The cartoon schematics in Fig. 2a show the evolution of primary instability patterns derived from experimental results. We found the primary wrinkling appeared at a strain of 0.67% with an average wavelength of ~1.2 μm. As the shear strain reached 1.5%, the wavelength decreased to approximately 360 nm. Then, we unloaded and observed that when the strain regressed to 1.1%, the wavelength returned to around 500 nm. This observation indicates that the primary instability should be completely recoverable. The experimentally measured profiles from monolayer 2D materials were shown in Fig. 2b [3D morphology of the wrinkling structure was scanned by atomic force microscopy (AFM); see details in Method], where the deflection and the lateral position were normalized by the height and the width of wrinkle, respectively. We define that $\omega$ as the coordinate along the sheet measured from the short axis and $\psi$ as the coordinate from the long axis. The normalized deflection profiles collapse onto one curve as predicted by the classical membrane analysis: $\frac{z}{h} = \sin\left(\frac{\pi\omega}{\lambda}\right)$, where $z$ is the out-of-plane deflection profile, $h$ is the wrinkle height (deflection at the center), and $\lambda$ is the half-wavelength from the wrinkle center. The morphology of the wrinkle along the long axis can also be approximated by trigonometric functions: $\frac{z}{h} = \sin\left(\frac{\pi\psi\sin\beta}{W}\right)$, where $\beta$ is the orientation of wrinkles (the statistic values of $\beta$ have been shown in Fig. 2c and Supplementary Fig. 7). Since the wrinkling regions are subjected to tensile and compressive stresses along the major and minor axes, respectively, we simplify the instability of 2D materials to the wrinkling behavior of a rectangular sheet with length $W/\sin\beta$ and width $2\lambda$. Then, we assume the out-of-plane morphology of wrinkles follows the equation: $z(\omega,\psi) = h\sin(k\pi\omega)\sin(m\pi\psi)$, $k = \frac{i}{\lambda}$, $m = \frac{j\sin\beta}{W}$, $i$ and $j$ are positive integers representing the wavenumbers (Supplementary Fig. 8).

The relationship between strain, the geometry of wrinkles, and the material's properties could be determined through the minimization of system potential energy. Since the wavelengths in the experiment are over a hundred nanometers, it should be reasonable to treat the 2D materials as continuums[33]. The detailed analysis can be seen in Supplementary Note 1 and Supplementary Figs. 8 and 9. The total elastic energy $U_{tot}$ of the wrinkling system comes from the bending $U_b$ and stretching $U_s$ of the 2D material, that is, $U_{tot} = U_b + U_s$. The bending energy per unit area is given as:

$$U_b = \frac{D}{2} \int\int \left(\nabla^2 z\right)^2 d\omega d\psi \qquad (1)$$

Here, $D$ is bending stiffness. Then stretching energy per unit area including the in-plane shear and the out-of-plane deflection in the plate structure which can be obtained as:

$$U_s = \frac{1}{2} \int\int \left[\sigma_\omega \left(\frac{\partial z}{\partial \omega}\right)^2 + \sigma_\psi \left(\frac{\partial z}{\partial \psi}\right)^2 + 2\tau_{\omega\psi} \left(\frac{\partial z}{\partial \omega}\frac{\partial z}{\partial \psi}\right)\right] d\omega d\psi \qquad (2)$$

where $\sigma_\omega$, $\sigma_\psi$, and $\tau_{\omega\psi}$ represent compressive, tensile, and shear stresses, respectively. Here, two main issues should be addressed: the essential condition for instability to arise and the geometric dimensions of the wrinkle once instability has occurred.

We consider that the condition for instability to arise is that there is a non-zero solution for $\frac{\partial U}{\partial h} = 0$, so that it could be transformed into an eigenvalue problem. Firstly, we consider the case ($i = j = 1$) that there is a single wrinkle both horizontally and vertically, corresponding to the primary instability stage. We simplify the analysis by assuming that the

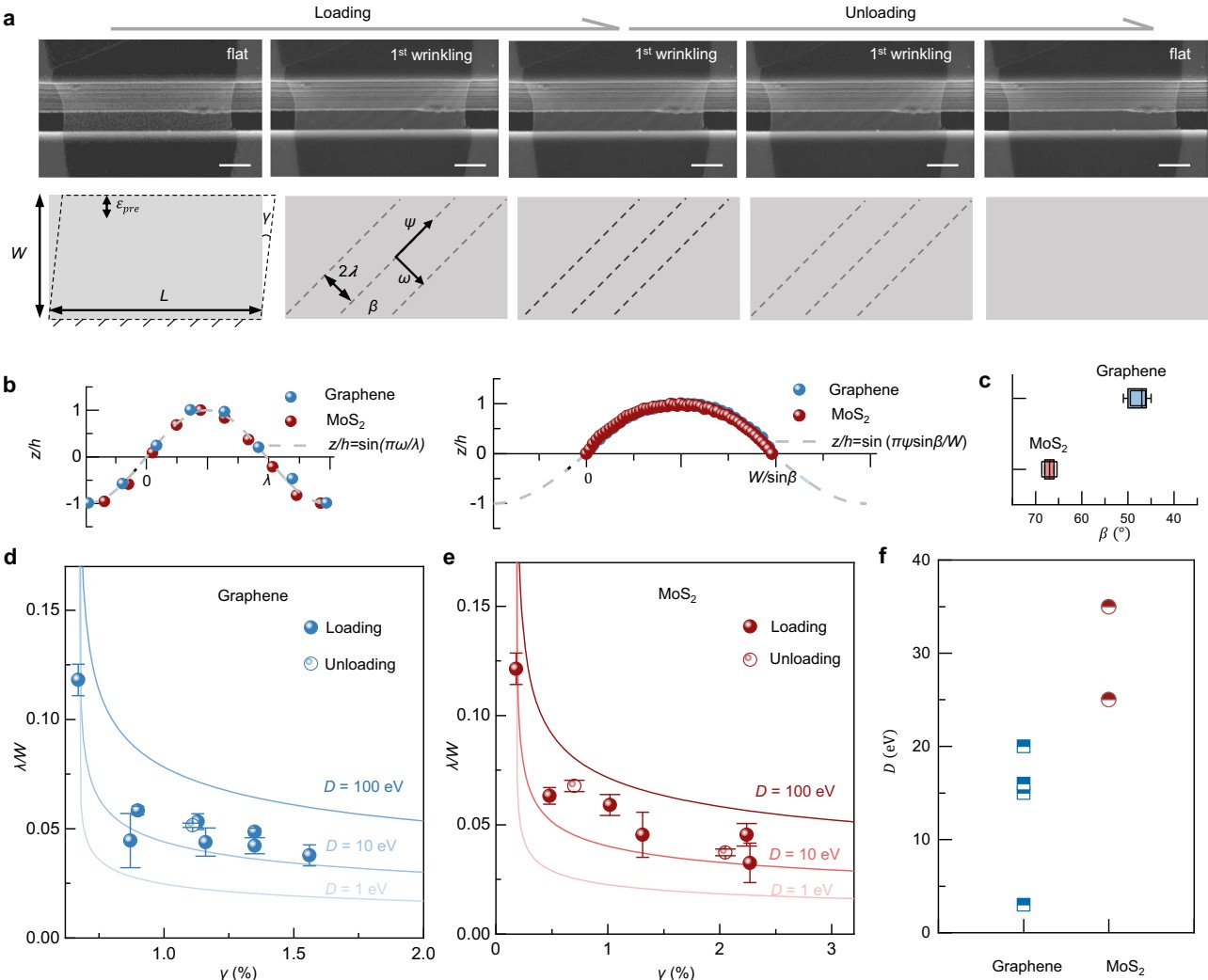

**Fig. 2 | Primary instability and bending stiffness measurement. a** In situ images illustrate the process of primary instability and recovery of monolayer graphene, where the cartoon schematics show the geometrical characteristic and evolution of wrinkling structure. **b** Normalized deflection profiles measured from two principal directions of monolayer graphene (blue markers) and MoS$_2$ (red markers) wrinkles which show the sinusoid shapes with different periods. **c** The statistical results show the angles between graphene ($\beta \sim 45°$) and MoS$_2$ ($\beta \sim 68°$) wrinkles and the horizontal direction, respectively. Error bars in **c**–**e** represent the standard deviations of measured angles of 5 wrinkles. **d**, **e** The curves show the theoretical normalized wrinkling wavelengths versus shear strains with different bending stiffness of monolayer graphene (The basic parameters were set as 2D Modulus $Et = 340$ N/m; Poisson's ratio $v = 0.165$; Prestrain $\varepsilon_{pre} = 0.172\%$; Sample width $W = 4770$ nm) and MoS$_2$ ($Et = 170$ N/m, $v = 0.27$, $\varepsilon_{pre} = 0.28\%$, $W = 4770$ nm), where solid and hollow balls indicate the data collected from instability and recovery process in experiments, respectively. Error bars represent the standard deviations of measured wavelength data from 5 wrinkles. **f** The scatter plots show the bending stiffness of monolayer graphene and MoS$_2$ which were counted from six samples. Scale bars: **a** 2 µm.

wrinkles release the normal compressive stress, which is compatible with the concept in tension field theory[34], because flat 2D materials cannot sustain compressions. The pretension strain versus primary critical shear strain can be approximated as $\varepsilon_{pre} \sim \frac{\gamma^{cr1}\sqrt{v}(1-v)}{2v}$, where $v$ is Poisson's ratio. The detailed analysis can be found in Supplementary Note 1 and Supplementary Fig. 9. The critical stress of the primary instability is written as

$$\sigma_\omega^{cr1} \sim - \left( \frac{\pi^2 D}{\lambda^{cr1^2} t} + \frac{\lambda^{cr1^2} \sin \beta^2}{W^2} \frac{E \varepsilon_{pre}}{1-v} \right) \qquad (3)$$

Therefore, the critical stress when the primary instability occurs is related to primary critical wavelength, bending stiffness, and pretension, where the critical strain, wrinkling wavelength and orientation $\beta$ are experimentally measurable. Furthermore, through the minimization of potential energy by wavelength, $\frac{\partial U}{\partial \lambda} = 0$, we obtain that,

$$\frac{\lambda}{W} \sim \left\{ \frac{2D\pi^2(1-v^2)}{EW^2 t \sin \beta^2 [-(1+v)\varepsilon_{pre} + (1-v)\sqrt{\varepsilon_{pre}^2 + \gamma^2}]} \right\}^{\frac{1}{4}} \qquad (4)$$

By Eq. 4, the wrinkling wavelength is related to bending stiffness, Young's modulus, pretension, and shear strain (the similar scaling law was also derived by the equilibrium equation[35,36]). Indeed, contrary to the well-known planar properties of 2D materials [e.g., Young's modulus of graphene (~340 N/m)[37] and MoS$_2$ (~170 N/m)[38] have been well measured], the measurement of weak bending stiffness of monolayer 2D materials is still extremely challenging, and previous bending results exhibit a large difference ranging from 1 eV to 10$^4$ eV (measured by nanoscale morphologies[39], electrostatic actuation[40], blister[41], and laser method[11], etc.). We here propose an approach, that

is, extracting the bending stiffness from the primary wrinkling process. Figure 2d, e shows $\lambda/W$ as a function of $\gamma$ for graphene and MoS$_2$, respectively, in which the solid lines reflect the theoretical solutions of three bending stiffnesses (1 eV, 10 eV, and 100 eV) with the fixed pretension. The theoretical wavelengths with varying pretensions are shown in Supplementary Fig. 9. Solid/hollow balls (we quantified two samples with equal pretensions for each material) represent experimentally collected data throughout the loading–unloading process. The overall trend of wavelength variation with shear strain follows the theoretical prediction: as the shear strain increased, the wavelength decayed fast and subsequently flattened. By fitting the experimental data with Eq. 1, Fig. 2f shows the bending stiffness of monolayer graphene and MoS$_2$ measured from six samples (Supplementary Fig. 10), where the average results are 3–20 eV and 25–35 eV, respectively. The results are higher than the value anticipated in atomic-scale measurement (~1 eV)[39]. Here we consider the effect of measurement error and initial corrugations on bending stiffness measurements, and the detailed analysis can be seen in Supplementary Note 2 and Supplementary Figs. 11–14. We infer that the major source of bending stiffness fluctuations is randomly distributed initial corrugations, which is similar to that thermal fluctuations and static ripples are expected to significantly stiffen micron-sized monolayer 2D materials[11]. According to theoretical calculation[42], the effective bending stiffness $D_{eff}$ of the thin film with corrugations $\langle A^2 \rangle$ can be written as $D_{eff}/D_0 = k_B T W^2 / 16 \pi D_0 \langle A^2 \rangle$, where $k_B T$ is the thermal energy, and $D_0$ is the intrinsic bending stiffness of 2D materials. Correspondingly, if the initial corrugations are 5 nm to 15 nm. the effective bending stiffness $D_{eff}$ would be about 5 eV to 47 eV, which roughly agrees with the range of experimental measurements. These measurement results could fill in part of the gaps between the nano-to-microscale bending of monolayer 2D materials[43], which is promising for evaluating the flexural resistance of 2D materials in device-scale applications.

## Secondary instability and stepwise wrinkle splitting

Upon the shear strain increased to a critical level after primary instability, through experimental observation (Fig. 3a and Supplementary Movie 1), we found the wrinkling splitting was accompanied by a halving of the wavelength from the end of the wrinkles (also named as wrinklon[19]), then gradually expanded throughout the entire wrinkle. When the strain is unloaded to zero, the wrinkling structure regresses to a completely flat state, which demonstrates that the secondary instability is also elastically recoverable. To visualize the detailed process, Fig. 3b shows the zoom-in region of local wrinkling splitting, and a series of cartoon diagrams were used to restore the subtle structural changes from the SEM images. Different from the instantaneous snap-through of classic plate buckling[44], our experiment showed a slower splitting process that is reminiscent of the growth of wrinklon, and eventually ended with two smaller but self-similar wrinkles. Such a process is akin to a binary-tree-like splitting[18], which appears to recur numerous times within a larger shear loading (Supplementary Movie 1). However, the unloading images in Fig. 3c show that the recovery path exhibited the gradual smoothing of wrinkling amplitude without pronounced wrinkling merging, in contrast to the splitting during the loading stage. To fully understand this nontrivial loading-unloading process, there are two key questions that need to be answered: what is the critical condition for secondary instability, and what are the key factors that dominate instability and recovery trajectories?

For the first issue, we assume that the conditions for the onset of secondary instability correspond to two wrinkling structures (the wavenumbers are labeled as $i_1, j$ and $i_2, j$) under the same stress state, $\sigma_\omega|_{i_1, j} = \sigma_\omega|_{i_2, j}$. The detailed analysis in Supplementary Note 3 implies

that the normalized critical stress of the secondary instability can be determined by,

$$\bar{\sigma}_\omega^{cr2} = \frac{W^2 t \sigma_\omega^{cr2}}{\pi^2 D \sin \beta^2} = \left(1 + \bar{\lambda}^{cr2 2}\right)^2 \left(1 - \eta \bar{\lambda}^{cr2 2}\right)^{-1} \tag{5}$$

Here, two principal stresses are assumed to have a proportional relationship: $\eta \sigma_\omega = \sigma_\psi$, $\eta \sim (\nu, \varepsilon_{pre}, \bar{\lambda}^{cr1})$. Therefore, by knowing the stress ratio $\eta$ through primary critical wavelength ($\lambda^{cr1}$) and strain ($\gamma^{cr1}$), we obtain the critical wavelength ($\lambda^{cr2}$), strain ($\gamma^{cr2}$), and stress ($\bar{\sigma}_\omega^{cr2}$) of the secondary instability. In other words, we can modulate the secondary instability patterns by quantitatively changing the shear strain. The procedure has been interpreted in Supplementary Note 3 and Supplementary Fig. 15.

In Fig. 3d, we show both theoretically (lines) and experimentally (balls) measured wavelength versus shear strain with the known pretensions. As shear strain increased during primary instability, the experimentally recorded wavelengths decreased (light blue balls), and the reduction gradually slowed, which follows the prediction by Eq. 1 (solid line). Nevertheless, as the strain reached about 3%, the downward trend of wavelength intensified, which means the emergence of secondary wrinkling (also refer to Fig. 3b and Supplementary Movie 1). We found the wavelength didn't decrease by half instantaneously (the dashed line predicts the wavelength variation after instantaneous wrinkle splitting), instead decreased in a roughly linear fashion (dark blue balls). Combined with the schematics in Fig. 3b, this abnormal decreasing trend of wavelength comes from the wrinkling splitting rather than the sudden bifurcation. The mechanical interpretations of such stepwise wrinkle splitting can be seen in Supplementary Fig. 16. Another evidence of the stepwise instabilities was the stress-strain curve in Fig. 3e: The stress change was reasonably smooth during the first instability stage, but the stress had several aberrant sudden drops after entering the second instability stage. Compared with the zoom-in stress curve and the inset SEM images in Fig. 3e, we found the process of wrinkle splitting was accompanied by sudden changes in stress. Overall, we clarified the critical stresses of such stepwise instabilities, but it remains unclear for the second issue about different paths in the loading-unloading process which may be related to the nontrivial stress drops.

## Instability and recovery paths

Here, we explored the evolution of the instability patterns of monolayer graphene and their underlying mechanisms using large-scale molecular dynamics (MD) simulations (Modeling details can be found in Method and Supplementary Fig. 17). Monolayer graphene was created and subjected to in-plane shear deformation by imposing relative displacements on its bottom and upper edges[45] (Fig. 4a). Here, we mainly discuss the secondary instability since primary instability is trivial and straightforward to understand. Figure 4b shows the snapshots on initialization, generation, and propagation of the second instability in monolayer graphene. As the shear strain increased, wrinklons nucleated from both ends of the wrinkles ($\gamma = 4.0\%$), then propagated to the middle and separated wrinkles ($\gamma = 4.5\%$), and eventually favored a complete splitting of the wrinkles ($\gamma = 4.8\%$). During unloading, the secondary wrinkles did not merge but instead tended to reduce in amplitude until they smoothed completely. These simulation results not only recurred the experimentally observed reversible instabilities but also demonstrated the different instability and recovery trajectories (Fig. 3b and Supplementary Movie 4). The potential energy variation and shear stress-strain curves (Fig. 4c) are coincident during loading and unloading, exhibiting reversibility at the system level. Figure 4d shows the variation of half-wavelengths of wrinkles (where the hysteresis loop manifests the different instability and recovery trajectories), and the profiles of wrinkles are shown in Supplementary Fig. 18. The conflict between the reversibility in system

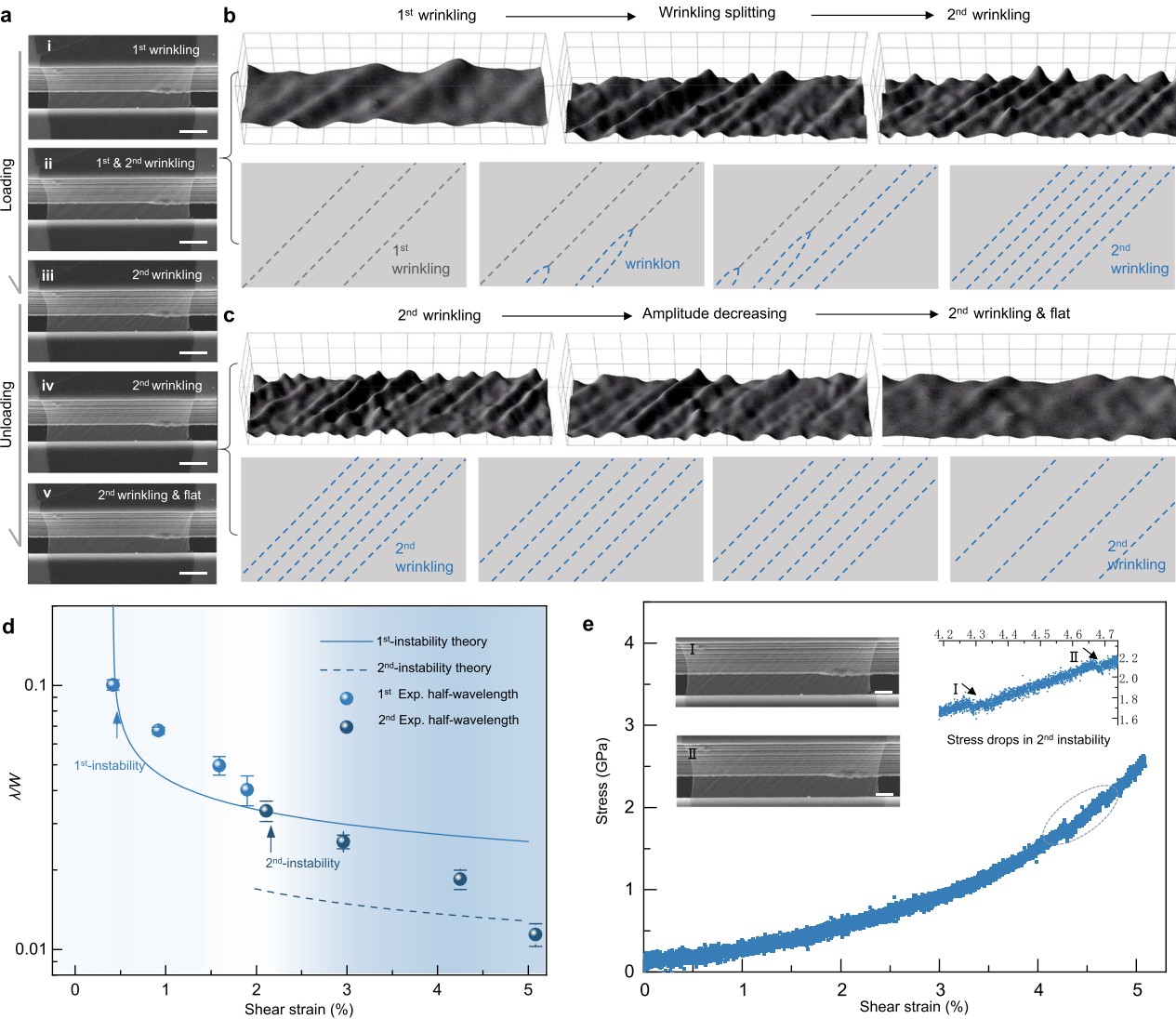

**Fig. 3 | Secondary instability and stepwise wrinkle splitting. a** In situ images illustrate the process of secondary instability and recovery of monolayer graphene, where the cartoon schematics show the geometrical characteristic and evolution of wrinkling structure. **b** The 3D morphologies show the loading path which corresponding to 1st wrinkling−splitting−2nd wrinkling, where the cartoon schematics denote the detailed process. **c** The 3D morphologies show the unloading path which corresponding to 2nd wrinkling−amplitude decreasing−2nd wrinkling, where the cartoon schematics denote the detailed process. **d** The curves show the theoretical normalized wrinkling wavelengths versus shear strains during 1st and 2nd instabilities of monolayer graphene (The graphene parameters were set as $Et = 340$ N/m, $D = 20$ eV, $v = 0.165$, $\varepsilon_{pre} = 0.172\%$, $W = 4770$ nm), where solid balls indicate the data measured in experiments, respectively. We used the light and dark blue to distinguish the instabilities. Error bars represent the standard deviations of measured wavelength data from 5 wrinkles. **e** The scatter plot shows the shear stress–strain curve of monolayer graphene. The inset curve was zoomed-in region as labeled by oval dashed box which indicates two stress drops during 2nd instability. The inset SEM figures show the wrinkling splitting with the drops of stress. Scale bars: **a** 2 μm and **e** 1 μm.

energy and the difference in instability-recovery paths can be explained by local stress redistribution.

We calculated the principal stress fields and their orientations (see Method and Supplementary Fig. 19 and Movie 5). Figure 4e exhibits the distribution of compressive principal stress before and after wrinklons nucleation, where the compressive stress was released by the out-of-plane deformation, resulting in substantially lower compressive stress in the middle of the wrinkle than that at either edge ($\gamma = 3.0\%$). As the shear strain increased ($\gamma = 5.0\%$), wrinklon formation initiated when the compressive stress reached the secondary critical stress ($\sigma_\omega^{cr2}$). The contour line of compressive stress exhibited a convex shape towards the center, as depicted in the right insets in Fig. 4e, which is verisimilar to the stress focusing in d-cone[46]. Furthermore, Fig. 4f shows the deflection profiles and compressive stress distributions along the dashed lines in Fig. 4e. The emergence of wrinklons (indicated by black

dashed boxes) was accompanied by a rise in compressive stress at both ends of the wrinkle. This phenomenon signifies the onset of wrinkle splitting, initiated once the stress reached $\sigma_\omega^{cr2}$. The wrinklon would then spread to the entire wrinkling region once the shear loading was increased. With the completion of wrinkling-splitting, the compressive stress accumulated by the wrinklon growth would return to a very low value ($\sigma_\omega \rightarrow 0$), indicating that the compressive stress has been released by the entirely split wrinkles (2nd wrinkles). As a result, once wrinkling-splitting is finished, the compressive stress in 2nd wrinkling region is always lower than $\sigma_\omega^{cr2}$, which means that the driving stress for wrinkling merging was lacking during the unloading stage. This compressive stress redistribution mechanism can be expanded to comprehend the underlying mechanics of numerous instability patterns of 2D materials[47,48], in addition to contributing to the explanation of the different instability and recovery trajectories.

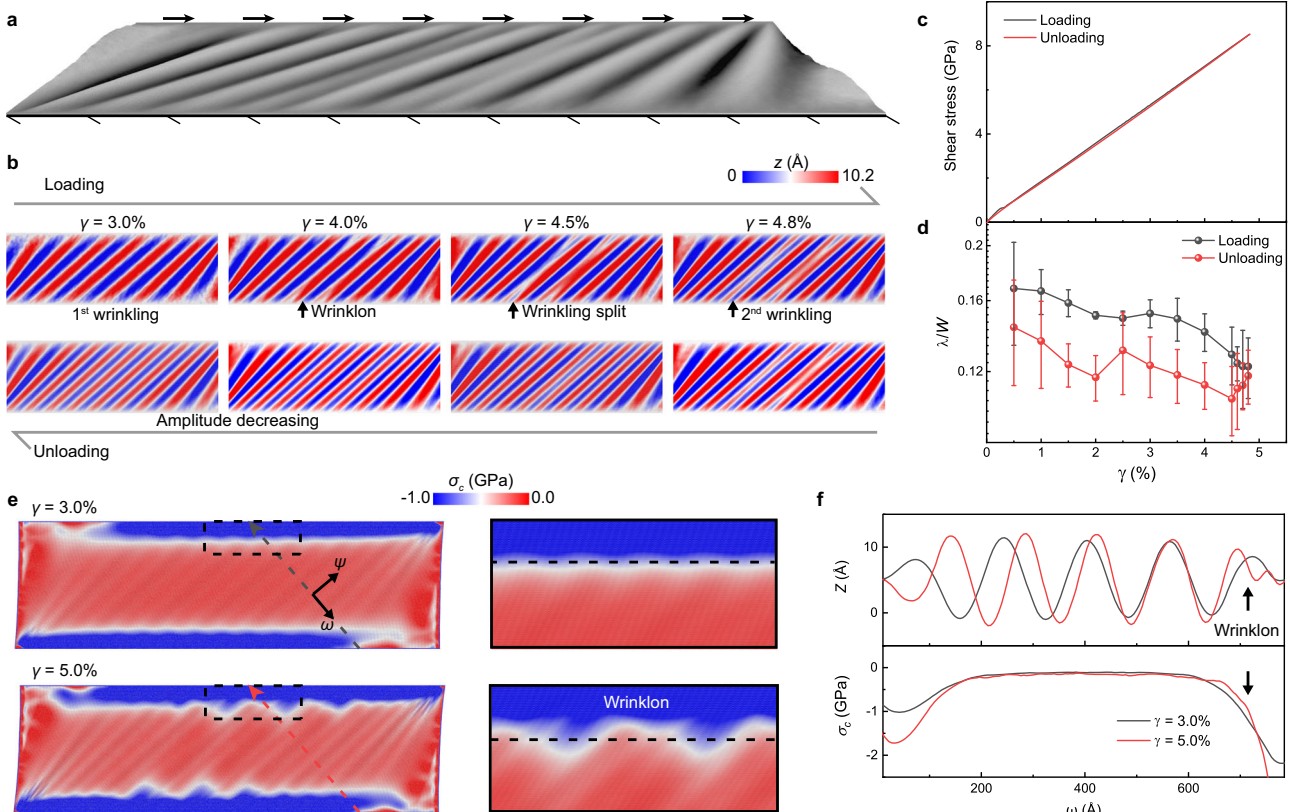

**Fig. 4 | Comparison of the loading and unloading processes in MD simulations.** **a** Schematic of in-plane shearing of monolayer graphene. **b** The diagrams show the loading and unloading paths. **c** The average shear stress-strain curve and **d** wavelengths versus shear strains during loading and unloading process. The error bars in **d** were obtained by calculating the average of multiple half-peak widths of the profile. **e** The minimum principal stress fields under different shear strains. Right insets show the zoom-in region of principal stress distribution in the left dashed boxes. **f** Deflection profiles and minimum principal stress distribution along $\omega$ direction at cross section indicated by dashed lines in **e**.

## Discussion

Our findings reveal a facile in situ shear technique to control the instability and provide a whole-process perspective of the stepwise instabilities and their reverse process in suspended monolayer 2D materials. We quantitatively investigated the critical stresses/strains of the stepwise instabilities and established a correlation between wrinkling geometry and strains. As a result, the instability morphology of suspended 2D materials can be designed based on a simple scaling law, which also works for other atomically thin films. The different instability and recovery paths of 2D materials are further disclosed, which are governed by changes in wrinkling wavelength and amplitude, respectively. In combination with MD simulations, we attribute the different paths to the local stress redistribution in monolayer 2D materials. The successful development of in situ in-plane shearing offers an experimental mechanical paradigm for evaluating the instability and bending properties of ultrathin materials. Even though we have primarily focused on the mechanics of the instability process, we believe that the local stress/strain and curvature associated with the instability process will be more crucial in physical and chemical applications, such as the potential to control proton transport[16] and tune electronic structure[7] by modulating the wrinkling morphology, as well as contribute to research on suspended 2D materials-electrodes[12]. Our research is anticipated to advance knowledge of the instability of atomically thin materials and the use of innovative suspended 2D material-based electronics and devices.

## Methods
### Sample preparation and characterizations
Natural graphite crystals and CVD-grown monolayer $MoS_2$ flakes were obtained from Shanghai Onway Technology Co., Ltd. The graphene

flakes used in the experiment were obtained by tape exfoliation of the natural graphite crystal, and the monolayer $MoS_2$ was used as received. A thin PMMA stamp, formed via spin coating, was hot pressed onto the flake as a supporting layer. This stamp could be detached from the Si substrate by wedging it in deionized water. The PMMA/graphene stack was then transferred to the MMD and accurately positioned under an optical microscope using a micromanipulator probe with a tungsten needle. A thin water layer was maintained between the stack and the MMD during alignment to ensure smooth movement and prevent material damage from direct contact[49]. Once the water layer evaporated, the graphene flake was deposited onto the desired gap of the MMD, leveraging the capillary effect for strong attachment to the substrate. An 80 °C annealing step was applied to further enhance adhesion. The final suspended graphene device was obtained by removing the PMMA supporting layer with acetone and using a critical point drying process. More detailed transfer procedures can be found in Supplementary Fig. 3. The Raman spectra were characterized by a WITec alpha300R confocal Raman microscope with a 532 nm laser, where the laser power was set as 1.7 mW to prevent thermal heating and damage of monolayer graphene and $MoS_2$. The morphology characterizations were performed by tapping mode (Dimension Icon, Bruker Company) to avoid disturbing the wrinkling structure.

### In situ nanomechanical test
The MMD with suspended graphene and $MoS_2$ samples were installed onto a SEM-incorporated Hysitron PI-85 nanoindentor for the push-to-shear test. A diamond indenter connected to a high-precision capacitive transducer was used to actuate the MMD and introduce a shear deformation to the sample, with an output of the load and

displacement data. A loading rate of 3 nm s$^{-1}$ was used, which corresponds to a shear strain rate of -0.06% s$^{-1}$. SEM video during the test was recorded to visualize the deformation of suspended nanosheets and the evolution of wrinkles. Note that the sample was tilted to a small angle (-20°) for better observation of the nanowrinkles. The shear strain of the sample was carefully measured through a digital image correlation method and the shear load on the sample was calculated by subtracting the inherent stiffness of the MMD[50]. The shear stress was calculated based on the sample width from the SEM image and a thickness of 0.34/0.65 nm for monolayer graphene/MoS$_2$.

## Molecular dynamic simulation

The large-scale molecular dynamics (MD) simulations were implemented using the large-scale atomic/molecular massively parallel simulator (LAMMPS)[51], and the interatomic interactions were described by the second-generation reactive empirical bond order (REBO) potential[52,53]. The monolayer graphene was imposed with free boundary conditions in three directions with the size of 1999.58 Å × 600.66 Å. After energy minimization using the conjugate gradient (CG) algorithm, the monolayer graphene was fixed with the upper and bottom edges and relaxed at 10 K for 100 ps under a canonical (NVT) ensemble, as shown in Fig. 4a. The width of the upper and bottom edges was set as 5 Å. The upper edge was moved at a constant velocity of 5 m/s with a thermal relaxation time of 2 ps. The Open Visualization Tool (OVITO) was used to visualize and capture atomic configurations[54]. The detailed simulation and stress visualization can be found in Supplementary Figs. 17–19.

## Data availability

The data that supports the findings of the study are included in the main text and supplementary information files. Raw data can be obtained from the corresponding author upon request. Source data are also provided as a Source Data file with this paper.

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

## Acknowledgements

This work was supported by the NSFC/RGC Joint Research Scheme (N_HKU159/22); National Natural Science Foundation of China (12232016,12202431); Shenzhen-Hong Kong-Macau Technology Research Program (Type C, SGDX20201103093003 01) and Research Grants Council of the Hong Kong Special Administrative Region, China under grant RFS2021-1S05. The numerical calculations in this paper have been done on the supercomputing system in the Supercomputing Center of the University of Science and Technology of China. Y.H. and Z.Z.H. appreciate helpful discussions with H.J.G.

## Author contributions

Y.H., J.Z.Z. and Y.L. conceived the idea and led the project. Y.H. and J.Z.Z. performed experiments. Y.H. conducted a theoretical analysis. Z.Z.H. conducted MD simulations. Y.H., J.Z.Z. and Z.Z.H. prepared the paper. Y.H., J.Z.Z., Z.Z.H., J.Z.C., M.Y.Z., H.A.W., and Y.L. discussed the results and revised the paper. Y.L. and H.A.W. supervised the studies.

## Competing interests

The authors declare no competing interest.
