## [Peer Review File · Nature Communications]

Tuning Instability in Suspended Monolayer 2D MaterialsREVIEWER COMMENTS

Reviewer #1 (Remarks to the Author):

The authors present an interesting study of shear-straining 1L graphene and MoS₂ via a push-to-shear device. The observations are, in general, worth and appealing to the community of 2D materials (and thin film) mechanics audience, the analysis and modelling as well. However, a crucial issue relates to a primary assumption that the stress transfer from the chip (SiO_x) to the monolayers is complete over the whole examined deflection range. I have serious concerns about this assumption, because achieving such strains on SiO_x would indeed be unprecedented. It comes to mind immediately when looking at the Figure 1e. The authors actually do not comment on this at all; they only note that, in the unloading stage, the decrease is continuous. While, towards the end of the manuscript, when discussing the asymmetry in the wrinkling-smoothing processes and modelling these two, the modelled stress-strain curves show no hysteresis at all, only the wavelength (and local stress) show hysteresis. From Fig. 1e I suspect the slippage might have accounted for about ~3.5% out of the indicated 5%.

Hence, before delving deeper into the analysis, the authors have to remove any doubts about possible slippage, otherwise the obtained values are incorrect, and the manuscript should not be published as it is.

There are few other issues noted below:

- the English has to be improved substantially – while in most of the manuscript, the level is OK, there are sentences which are difficult to understand due to missing prepositions etc.
- the authors should discuss the spread of values of D more. But it actually might be due to uneven slippage, as noted above, or due to uncertainty in determining the onset of wrinkling visually (from which the $\epsilon(\text{pre})$ is obtained)
- the description of the simulated wrinkle formation (page 13) should be rewritten; it is hard to follow now. E.g., it is unclear to what “red lines in dark regions” refers to, also Fig. 4e labels Wrinkle in the middle of the bottom-right panel, but in Fig. 4f, it is labelled at the edge
- in the final paragraph, the authors state “unprecedented” symmetry breaking – it is unclear why unprecedented

Reviewer #2 (Remarks to the Author):

This paper reports on in-situ measurements of shear induced wrinkling in graphene films, and develops models to understand the wrinkling behavior and instability transitions and extract material properties. These sorts of in-situ measurements are monumentally difficult to achieve and are well executed. The insights are important for making the connection between the atomic scale properties and >microscale device behavior. This reviewer particularly appreciates the understanding of wrinkle motion in Figure 3. The results will be of interest to the broad nanoscience and 2D materials mechanics communities. This reviewer has some important suggestions on moderating claims, and bringing out additional insight on the nanoscale origin of bending stiffness that will require major revisions, then will recommend the paper for publication.

1. While there is plenty of novelty and value in in-situ measurement and instability analysis, the introduction and discussion/conclusion need to take more care on the claims of novelty. Shear induced wrinkling was observed previously and a similar scaling law proposed. Most relevantly, see Figure 1 of: <https://www.nature.com/articles/nnano.2009.191> (reference 17 in the manuscript)

And as the authors have pointed out in the main text, there are many measurements on bending stiffness and shear modulus. As a result, while the abstract is fine, in the intro, “here we show”

paragraph, and conclusions, the authors need to moderate their claims of novelty to reflect the existing literature and to articulate what is different from these previous results.

2. I think there is a missed opportunity on learning more about the nature of bending stiffness in 2D materials. As the authors point out, there is a discrepancy between the atomic scale bending stiffness (~ 1 eV) and the macroscale ($\sim 10^4$ eV) and even state "Our measurement results could fill in part of gaps between the nano-to-microscale bending of monolayer 2D materials, which is actually promising for evaluating the flexural resistance of 2D materials in device-scale applications." But in the end, they shy away from actually investigating the origin of the values that they measure. In particular, the range in measured bending stiffness of 3 eV to 20 eV is enormous, especially given that the fit appears to be quite good on each value. This suggests that the variation comes from the initial fluctuation in the morphology rather than experimental error. The value of this in-situ measurement is the authors can actually connect the nanoscale initial corrugations to the resulting measured bending stiffness. This reviewer suggests the following to bring significant additional impact and achieve the stated goal of filling in the gap:

Each device has initial static height fluctuations, which could be indicative of inhomogeneous stresses. The authors should try to correlate the initial height fluctuations to the different measured bending stiffness? This reviewer understands that these are in arbitrary units, but the wrinkles could be used to calibrate the magnitude of the fluctuations.

3. The authors show loading and unloading curves on a total of 6 of different devices, but do not show two measurements from the same device. How repeatable are these measurements? E.g. if you re-load the device do you see the same wrinkle pattern emerge, and the same stress-strain relation? This could be used to get additional statistics on the material properties – e.g. by cycling the load multiple times, the initial wrinkling changes. Do the resulting measured material parameters remain consistent, randomly change or systematically trend through wear? If they change, the extracted material parameters could be correlated with the initial static fluctuations between each loading cycle as discussed in the previous point. If they remain similar, then it provides more reliable error bars on the material properties.

Minor comments:

1. In Figure 2, it is inappropriate to show a mean and error on 4 graphene and 2 MoS₂ devices, especially given the high spread. It would be better to just show a scatter plot of the individual values with the error for each device.

2. Confusingly, the introduction, abstract and conclusion contain numerous grammatical issues, while the main body of the text is comparatively well written. Edit for grammar.

Reviewer #3 (Remarks to the Author):

The manuscript by Hou et al. reports experimental results for wrinkle formation in graphene and MoS₂ monolayer suspended over a trench, which is part of the device. Through a "push to shear" delicate mechanical process carried out under SEM, the authors observe formation of periodic wrinkles followed by the formation of a secondary wrinkle pattern of smaller wavelength. The mechanism of formation is wrinkle splitting, as observed also in the reported molecular dynamics simulations. Under release, graphene flattens directly without revisiting the primary wrinkle pattern of larger wavelength. While I believe that these results merit publication in some capacity, the manuscript, as a whole, does not possess the caliber required for acceptance in Nature Communications. Some of the reasons are listed below:

1) The introductory part fails to be clearly articulate the importance of this work. Without a convincing explanation, the authors assert exaggerations without providing substantiating evidence: "Such binary

instability, as a basic feature of compressed atomically thin films, is deemed to play a vital role in physical properties of 2D materials." But authors do not elaborate why. The introduction is also written in a confusing manner and some paragraphs do not have a logical flow: "Besides, the static binary instability patterns, referring to the initial wrinkles could bifurcate into period-halving wrinkles, were experimentally discovered in few layers graphene and other membranes." "We achieve in a shear-induced reversible binary instability (include primary and secondary instabilities) of monolayer 2D materials, which can be quantified by a series critical stresses that governed by pretension, geometry and modulus of materials." Additionally, some concepts are not properly used and this again makes the manuscript confusing and difficult to read: "Is the restabilization process symmetric with the instability process?" Probably what the authors want to say here is "After unloading, does the recovery process backtrack the forward process?" if so "restabilization" and "symmetric" are not the proper words. As the manuscript eventually clarifies, the answer is no. However, it is improper to call "symmetry breaking" the fact that the recovery process does not backtrack the forward process. Similarly, referring to wrinkle splitting later on as "bifurcation" is also confusing.

2) The values of the elastic constants extracted from this experiment are in very modest agreement with what is expected from the previous literature. The shear modulus of 70 GPa is very low compared with what is reported in the experimental reference 29. The measured bending constants are large compared with theoretical predictions and other experiments. The provided explanation "this is expected given that thermal fluctuations and static ripples are expected to significantly stiffen micron-sized monolayer 2D" is not clear or convincing.

3) I find it a stretch calling a "paradox" the fact that the recovery process does not follow the same path as the forward process. Of course, upon external decompression, the wrinkles should flatten directly – a paradox would be if they would instead localize into the larger amplitude and larger wavelength wrinkles. Also the forward cause of the secondary wrinkle is not different from the formation of the primary one, i.e. compression release into bending! This should not be presented as a surprise, as the current manuscript does.

4) The scaling law derived from the continuum analysis seems to have some similarity with the continuum analysis of Cerda and Mahadevan in Phys. Rev. Lett. 90, 074302 . perhaps the authors could have a look at this paper to make better contact with previous understanding.

We gratefully thank the reviewers for the time and insightful comments to improve the quality of our manuscript. Provided below is our detailed response to each question.

Response to comments of Reviewer #1:

Q1: The authors present an interesting study of shear-straining 1L graphene and MoS₂ via a push-to-shear device. The observations are, in general, worth and appealing to the community of 2D materials (and thin film) mechanics audience, the analysis and modelling as well.

A1: We greatly appreciate your enthusiastic assessment and positive comments for our work. These comments are helpful to improve the quality of our manuscript.

Q2: However, a crucial issue relates to a primary assumption that the stress transfer from the chip (SiO_x) to the monolayers is complete over the whole examined deflection range. I have serious concerns about this assumption, because achieving such strains on SiO_x would indeed be unprecedented. It comes to mind immediately when looking at the Figure 1e. The authors actually do not comment on this at all; they only note that, in the unloading stage, the decrease is continuous. While, towards the end of the manuscript, when discussing the asymmetry in the wrinkling-smoothing processes and modelling these two, the modelled stress-strain curves show no hysteresis at all, only the wavelength (and local stress) show hysteresis. From Fig. 1e I suspect the slippage might have accounted for about ~3.5% out of the indicated 5%. Hence, before delving deeper into the analysis, the authors have to remove any doubts about possible slippage, otherwise the obtained values are incorrect, and the manuscript should not be published as it is.

A2: Thank you for your valuable comments. We do agree with you that possible boundary slippage can have a significant impact on the results of experimental measurement. Here, we added more experimental and theoretical evidences to address your concerns.

1) *Possibility of boundary slippage*

Experimental evidences

We first compared the SEM images of two samples before and after shear loading, respectively. As shown in Fig. R1(a), three impurity particles immobilized on the upper right, lower right, and lower left regions on the substrate were labeled to be Markers 1, 2, and 3, respectively. The distances of these markers from their nearest neighboring graphene edges are 656 nm, 273 nm, and 938 nm, respectively. The horizontal distance between

Marker 1 and Marker 2 is 742 nm. When the bottom Si substrate was displaced horizontally relative to upper Si substrate through shear loading, the distance between Markers 1 and 2 was measured as 992 nm. By comparing Figs. R1(a) and (b), it can be seen that only the distances between Markers 1 and 2 changed, while the distances between three markers and their nearest graphene edges remains invariant. Similarly, we obtained the same conclusion from the comparison in Figs. R1(c) and (d). Considering that there is no relative displacement between the marker and the substrate, our results demonstrate that there is no significant slippage of the graphene with respect to the substrate.

Fig. R1. Comparison of relative displacements between substrate and graphene layers. (a-b) SEM images of sample 1 before and after shear loading. (c-d) SEM images of sample 2 before and after shear loading.

Theoretical analysis

Secondly, the key parameter controlling the boundary slippage is the interfacial shear resistance between 2D material and silicon substrate. Previous studies have shown that the shear resistance between 2D material and silicon substrate is about 1~2 MPa, which is much higher than that between 2D materials, and thus it can be regarded as a strong interface [*Nano Lett.*, 2013, 13(6), 2605-2610; *Phys. Rev. Lett.*, 2017, 119(3), 036101; *Phys. Rev. Lett.*, 2018, 121(26), 266101]. In addition, since the surface of PTS is not atomically smooth during the preparation process, defects and morphological undulations on the silicon surface can also have the pinning effect on the 2D material. Here, we estimate for the possibility of sliding of monolayer graphene on the substrate. According to previous work (*Compos. Sci. Technol.*, 2017, 149: 220-227), the critical sliding stress can be described by the shear-lag model, given by $\sigma_s = \tau_{cr}L/t$, where L is the overlap length, t is the graphene thickness (~ 0.34 nm). The shear strength τ_{cr} between graphene and the substrate is about 1~2 MPa. Using the smallest overlap length in Fig. S2(a) ($L \sim 1.62$ μm), the critical sliding stress $\sigma_s > 4.76$ GPa, which is higher than the maximum principle stress during push-to-shear process. Consequently, the sliding of monolayer graphene on the substrate cannot occur in our testing samples.

Fig. R2. (a) Typical testing sample and its geometric parameters in experiment. (b) Equilibrium analysis of monolayer graphene on the substrate.

2) Explanation for hysteresis in stress-strain curves

Fig. R3. The original loading-unloading curve of the sample in Fig. 1e in manuscript. In our experiments, PTS MMD is fixed to a metal holder by a conductive carbon adhesive tape due to its inherent damping properties, which serve to mitigate the instrument-induced vibrations. Fig. R3 shows the original force-displacement curves during loading-unloading, where the stress-strain curves of the sample can be extracted by subtract the stiffness of PTS. When PTS was loaded by diamond indenter pushing, a hysteresis in the force-displacement curve occurs at large strains because of the carbon adhesive tape has viscoelasticity. It should be noted that while the viscoelasticity of the adhesive tape can potentially influence stress measurements at larger strains, our emphasis in this study is on the evolution of shear strain versus wavelength. Therefore, the hysteresis curve in stress-strain relationship will not affect our analysis of instability behavior.

3) Effect of slippage (MD simulations)

Fig. R4. The wrinkle diagrams of monolayer graphene at shear strain $\gamma=2\%$ under different spring constraints of (a) 34 N/m, (b) 340 N/m, (c) 3400 N/m, and (d) rigid fixed.

The effect of edge slippage and deformation of monolayer graphene can be described using the push-to-shear model with edge spring constraints. As shown in Fig. R2(a), the overlap area of the left edge is smaller than that of the right edge, leading to that the sliding of graphene tends to occur on the left edge. Here, we only consider that the monolayer graphene is bounded by a spring on one side. The spring stiffness determines the ability of the edge to deform or slip, where a small stiffness indicates a large edge deformation or easy to slip. As shown in Fig. R4, we use MD simulations to illustrate the effect of edge constraints. With the increase of boundary constraint, the wavelength of wrinkle decreases and the wavenumber increases, which is similar to the effect of prestrain. From the shear-lag model, we obtain the displacement of edge is $u = \sigma_s^2 t / 2E\tau_{cr}$, where E is the in-plane modulus of graphene. Then, the equivalent constraint stiffness of the edge is given by $K = 2EW\tau_{cr} / \sigma_s$, where W is the width of samples. Substituting the parameters in Fig. R2(a), the equivalent stiffness is much larger than the in-plane stiffness of monolayer graphene, i.e., $K \gg Et$. As a result, both prestrain and wide sample contribute to restrain the effect of edge sliding and deformation.

These discussions have been added in Manuscript, Page 5, Lines 106-108 and Supplementary Materials, Page 6-7, Lines 88-110.

There are few other issues noted below:

Q3: the English has to be improved substantially – while in most of the manuscript, the level is OK, there are sentences which are difficult to understand due to missing prepositions etc.

A3: Thanks. We have revised and embellished the language of the manuscript, which were highlighted in the revised version.

Q4: the authors should discuss the spread of values of D more. But it actually might be due to uneven slippage, as noted above, or due to uncertainty in determining the onset of wrinkling visually (from which the epsilon(pre) is obtained)

A4: Here, we revisited the experiments and suggest several factors that may have an effect on the bending stiffness measurement which mainly include slippage, measurement error, initial corrugations and irradiations. More detailed experimental evidence and analysis of the effects of initial corrugations and irradiation is appended to the content of the Supplementary Materials as well as to the response in question 3 of Reviewer 2. We think the major source of bending stiffness fluctuations is randomly distributed initial corrugations. Here we mainly focus on your questions. Based on the analysis in A2, we conclude that edge slippage is not a major source of error in the bending stiffness measurements. Secondly we consider the effect of determining the onset of wrinkling visually. We recognize that there is an uncertainty in determining the onset of wrinkling. Here, the bending stiffness of 2D materials in our experiment can be expressed as

$$D \sim \frac{1}{2\pi^2(1-\nu^2)} \left(\frac{\lambda}{W}\right)^4 Et W^2 \sin^2 \beta \left[-(1+\nu)\varepsilon_p + (1-\nu)\sqrt{\varepsilon_p^2 + \gamma^2} \right].$$

The error of bending stiffness originates from the measurement of λ , β , ε_p , and γ . Then, the error transfer of bending stiffness is given by

$$\left|\frac{\Delta D}{D}\right| = \left|\frac{4\Delta\lambda}{\lambda}\right| + \left|\frac{2\Delta\beta}{\cot\beta}\right| + \frac{\left|\Delta\varepsilon_p \left[-(1+\nu)\varepsilon_p + (1-\nu)\sqrt{\varepsilon_p^2 + \gamma^2} \right] + |\Delta\gamma\gamma(1-\nu)|\right|}{-(1+\nu)\varepsilon_p\sqrt{\varepsilon_p^2 + \gamma^2} + (1-\nu)(\varepsilon_p^2 + \gamma^2)}.$$

where $\Delta[\cdot]$ indicates the variation of parameters, and $\Delta[\cdot]/[\cdot]$ reflects the relative error. This equation shows the influence of relative error of each measurement quantity on relative error of bending stiffness. Since we obtain the bending stiffness by fitting the function of λ with respect to γ , the error from β and ε_p can be neglected, and only effect of relative error from the half wavelength λ is discussed below.

These discussions have been added in Manuscript, Pages 9-10, Lines 194-204 and Supplementary Materials, Page 16-19, Lines 208-298.

Q5 the description of the simulated wrinklone formation (page 13) should be rewritten; it is hard to follow now. E.g., it is unclear to what “red lines in dark regions” refers to, also Fig. 4e labels Wrinklone in the middle of the bottom-right panel, but in Fig. 4f, it is labelled at the edge

A5: Thank you for your helpful comments. In the revised manuscript, we rewrite the part of simulated wrinkle formation as “The emergence of wrinkles (indicated by black dashed boxes) was accompanied by a rise in compressive stress at both ends of the wrinkle. This phenomenon signifies the onset of wrinkle splitting, initiated once the stress reached σ_{ω}^{cr2} ”. The labels in Fig. 4 have been changed. (Manuscript, Page 13, Lines 277-284 and Page 26)

Q6: *in the final paragraph, the authors state “unprecedented” symmetry breaking – it is unclear why unprecedented.*

A6: Thank you for pointing out this inaccurate expression. Here, based on your comments as well as those of other reviewers, we have removed those words like “*unprecedented*” and “*symmetry breaking*”. Currently, we have revised the manuscript to “different instability and recovery process”. (Manuscript, Page 14, Lines 297-306)

Response to comments of Reviewer #2:

This paper reports on in-situ measurements of shear induced wrinkling in graphene films, and develops models to understand the wrinkling behavior and instability transitions and extract material properties. These sorts of in-situ measurements are monumentally difficult to achieve and are well executed. The insights are important for making the connection between the atomic scale properties and >microscale device behavior. This reviewer particularly appreciates the understanding of wrinkle motion in Figure 3. The results will be of interest to the broad nanoscience and 2D materials mechanics communities. This reviewer has some important suggestions on moderating claims, and bringing out additional insight on the nanoscale origin of bending stiffness that will require major revisions, then will recommend the paper for publication.

A1: We appreciate very much for your positive comments and endorsing our understanding of the wrinkling behavior of 2D materials. Following your comments, we supplemented more results to make the analysis and conclusion more comprehensive.

Q2. *While there is plenty of novelty and value in in-situ measurement and instability analysis, the introduction and discussion/conclusion need to take more care on the claims of novelty. Shear induced wrinkling was observed previously and a similar scaling law proposed. Most relevantly, see Figure 1 of: <https://www.nature.com/articles/nnano.2009.191> (reference 17 in the manuscript) And as the authors have pointed out in the main text, there are many measurements on bending stiffness and shear modulus. As a result, while the abstract is fine, in the intro, “here we show” paragraph, and conclusions, the authors need to moderate their claims of novelty to reflect the existing literature and to articulate what is different from these previous results.*

A2: Thank you for appreciating the novelty of this work. We reorganized the presentation about the content of this paper to highlight the differences from the literature, please see below for specific changes. (Manuscript, Pages 3-4, Lines 50-80)

“Experimentally, apply controllable out-of-plane deformation for creating and tuning the instability in suspended 2D materials is rather challenging. Previous work utilized thermal stress to achieve buckling compression on suspended graphene layers¹⁷, where the static periodic wrinkles and wrinklons (which referring to the initial wrinkles could split into period-halving wrinkles¹⁵) were observed in the buckled graphene. Similar phenomena were also discovered in other thin films^{16, 17, 18, 19}. However, the instability and recovery process of suspended monolayer 2D materials cannot yet be well quantified due to the absence of mechanical measurements. Besides,

“In this work, we implement in situ shear loading-unloading of suspended monolayer 2D materials that consist of graphene and molybdenum disulfide (MoS₂), by employing a mechanical push-to-shear (PTS) strategy, which allows precise tuning of the emergence and evolution of instability patterns. Compared to the realized micro/nano-scale mechanical experiments for 2D materials, e.g. tension, compression, and bending experiments (Fig. 1a)^{17,23,24}, the in-plane shearing can precisely control the instability behavior of thin film. In a broad sense, In our measurement, we observed the arise of primary and secondary instabilities under increasing shear loadings. The loadings can be quantified by a series of critical stresses governed by pretension, geometry, and modulus of materials.....”

Q3. I think there is a missed opportunity on learning more about the nature of bending stiffness in 2D materials. As the authors point out, there is a discrepancy between the atomic scale bending stiffness (~1 eV) and the macroscale (~10⁴ eV) and even state “Our measurement results could fill in part of gaps between the nano-to-microscale bending of monolayer 2D materials, which is actually promising for evaluating the flexural resistance of 2D materials in device-scale applications.” But in the end, they shy away from actually investigating the origin of the values that they measure. In particular, the range in measured bending stiffness of 3 eV to 20 eV is enormous, especially given that the fit appears to be quite good on each value. This suggest that the variation comes from the initial fluctuation in the morphology rather than experimental error. The value of this in-situ measurement is the authors can actually connect the nanoscale initial corrugations to the resulting measured bending stiffness. This reviewer suggests the following to bring significant additional impact and achieve the stated goal of filling in the gap: Each device has initial static height fluctuations, which could be indicative of inhomogeneous stresses. The authors should try to correlate the initial height fluctuations to the different measured bending stiffness? This reviewer understands that these are in arbitrary units, but the wrinkles could be used to calibrate the magnitude of the fluctuations.

A3: We thank the reviewer for pointing out the potential factors affecting bending stiffness. We, therefore, made the following assumptions and relevant experimental and theoretical analysis, as summarized below.

1) *No obvious boundary slippage*

In our experiment, we found that there is no boundary slippage (Specific analysis can be found in response A2 to Reviewer 1), so its effect on the bending stiffness measurements is negligible.

2) *Measurement error analysis*

Frankly, the measurement errors are inevitable in the nanoscale mechanical measurements. The bending stiffness of 2D materials in our experiment can be expressed as

$$D \sim \frac{1}{2\pi^2(1-\nu^2)} \left(\frac{\lambda}{W}\right)^4 Et W^2 \sin^2 \beta \left[-(1+\nu)\varepsilon_p + (1-\nu)\sqrt{\varepsilon_p^2 + \gamma^2} \right]$$

The error of bending stiffness originates from the measurement of λ , β , ε_p , and γ . Then, the error transfer of bending stiffness is given by

$$\left| \frac{\Delta D}{D} \right| = \left| \frac{4\Delta\lambda}{\lambda} \right| + \left| \frac{2\Delta\beta}{\cot \beta} \right| + \frac{|\Delta\varepsilon_p[-(1+\nu)\varepsilon_p + (1-\nu)\sqrt{\varepsilon_p^2 + \gamma^2}]| + |\Delta\gamma\gamma(1-\nu)|}{-(1+\nu)\varepsilon_p\sqrt{\varepsilon_p^2 + \gamma^2} + (1-\nu)(\varepsilon_p^2 + \gamma^2)}$$

where $\Delta[\cdot]$ indicates the variation of parameters, and $\Delta[\cdot]/[\cdot]$ reflects the relative error. This equation shows the influence of relative error of each measurement quantity on relative error of bending stiffness. Since we obtain the bending stiffness by fitting the function of λ with respect to γ , the error from β and ε_p can be neglected, and only the effect of relative error from the half wavelength λ is discussed below.

3) Effect of initial corrugations

As atomically thin films, suspended monolayers of 2D materials have been shown to have initial corrugations with undulations in the range of a few nm to a hundred nm [*Nat. Mater.*, 2007, 6(11), 858-861; *Nature*, 2015, 524(7564), 204-207]. The effect of these corrugations approximate an increase in the equivalent thickness of the 2D materials and therefore leads to an increase in bending stiffness. It should be noted, however, that in SEM experiments, the observation of the initial tiny corrugations is nearly impossible because SEM imaging cannot accurately detect height information. Here, we utilized the same transfer procedures as in this manuscript to transfer monolayer graphene onto the silicon substrates with through-hole arrays (in Fig. R5).

Fig. R5. Monolayer graphene before transfer (a) and transferred graphene on silicon substrate with through-hole arrays (b).

To avoid the influence of AFM tip on corrugations, we used the non-contact mode AFM to image the height profiles of suspended graphene. As shown in Fig. R6, we found that the graphene in the suspended region does have corrugations and these undulations are not significantly oriented. We counted the undulations of graphene in six overhanging regions

and found that the undulation heights ranged from a few nanometers to a dozen nanometers. The curves in Fig. R6 illustrates the height profile across the line-scanning in each region where the undulations in the three suspended regions ranging from a minimum of approximately 2 nm to a maximum of around 11 nm.

Fig. R6. AFM scanning results of suspended monolayer graphene on different holes and the height profile along the line scanning.

According to theoretical calculation (*J. Mech. Phys. Solids*, 2017, 107, 294-319), the effective bending stiffness D_{eff} of the thin film with corrugations $\langle A^2 \rangle$ can be written as $D_{\text{eff}}/D_0 = k_B T W^2 / 16\pi D_0 \langle A^2 \rangle$, where $k_B T$ is the thermal energy, and D_0 is the intrinsic bending stiffness of 2D materials. Correspondingly, if the initial corrugations are 5 nm~15 nm, the effective bending stiffness D_{eff} is 5 eV~47 eV, which roughly agrees to the range of experimental measurement. Additionally, the amplitude the initial fluctuation is restrained by the prestrain, given by $\langle A^2 \rangle = \frac{k_B T}{4\pi E \varepsilon_{\text{pre}}} \ln \left(1 + \frac{E W^2 \varepsilon_{\text{pre}}}{4\pi^2 D_0} \right)$. Due to the random prestrains, the initial fluctuations of our sample are not completely consistent, leading to different initial fluctuations in Fig. S10.

4) Possible effect of irradiations

According to the previous study, ion beam irradiation will have a significant effect on the bending stiffness of 2D materials. In our experiments, we selected samples with regular boundaries without the need for ion beam processing, so the effect of ion irradiation can be neglected. On the other hand, high-voltage electron microscopy, such as transmission electron microscopy electron beams, can damage 2D materials and cause defects, as well as affect bending properties. Here, we use a SEM voltage of 10kV, so we can guarantee that the 2D materials will not be damaged by the bombardment of high-energy electron beam streams. Besides, to avoid large amounts of carbon buildup during the test, we clean the electron microscope chamber by plasma before each test, thus minimizing the effect of

carbon buildup on the measurements. However, we also realized that, when the sample is exposed to prolonged periods within the electron microscope, amorphous carbon will accumulate, leading to increased values in subsequent bending stiffness measurements. This phenomenon occurs with a mechanism akin to that of 3).

These discussions have been added to Manuscript, Pages 9-10, Lines 194-203 and Supplementary Materials, Pages 16-19, Lines 208-298.

Q4. The authors show loading and unloading curves on a total of 6 of different devices, but do not show two measurements from the same device. How repeatable are these measurements? E.g. if you re-load the device do you see the same wrinkle pattern emerge, and the same stress-strain relation? This could be used to get additional statistics on the material properties – e.g. by cycling the load multiple times, the initial wrinkling changes. Do the resulting measured material parameters remain consistent, randomly change or systematically trend through wear? If they change, the extracted material parameters could be correlated with the initial static fluctuations between each loading cycle as discussed in the previous point. If they remain similar, then it provides more reliable error bars on the material properties.

A4: We highly appreciate these insightful comments. Despite being challenging to conduct such experiments; we supplemented more data to address your concern. In Fig. R7(a), we firstly present the unprocessed force-displacement curves of the same sample under different loading-unloading cycles. It is noteworthy that the forces align closely during both loadings, implying that the material's stress-strain response remains largely consistent with the increasing number of measurements. Note that, as the 3rd loading surpasses the strength of monolayer graphene, an abrupt decrease in force curve is evident in loading stage. This decline signifies shear damage to the material.

Besides, the SEM images captured in two cyclic loadings are shown in Fig. R7(b), both of which indicate the wrinkle-splitting phenomena at the similar shear strain. To further investigate the effect of multiple measurements on wrinkling patterns, in Fig. R8, we present the wrinkling behavior (1st instability) observed in the same sample subjected to three loading-unloading cycles. At small strains (<1%), the differences in wavelength are discernible among the three loadings. These findings suggest that, following every loading, there is a shift in the prestrains of the sample. This phenomenon is attributed to the fact that the unloading process does not fully restore the initial state attained after the last loading, consequently giving rise to residual strain. At larger shear strains, the relationship of wavelengths under the three loadings observed as $\bar{\lambda}_1 < \bar{\lambda}_2 < \bar{\lambda}_3$,

signifies that, as the number of measurements increases, the bending stiffness of the sample also experiences an incremental rise. In our previous analysis, we illustrated the significant impact of the initial corrugations on bending stiffness. The divergence in bending stiffness under various loadings arises, on one hand, from alterations in the initial corrugations after each loading cycle.

On the other hand, as such tests were mostly conducted under an electron microscope, prolonged exposure may result in the formation of amorphous carbon deposits on the material surface. These deposits may modify the initial corrugations and thus increase effective thickness of the sample. These factors also contribute to an upward trend in the bending stiffness with increased numbers of tests. Consequently, in the results section of our manuscript focusing on bending stiffness, we mainly present data from the initial test. It is noteworthy that this paper primarily delves into the investigation of material instability behavior. The proposed bending stiffness measurements serve as a methodological case, and further refinements and more precise measurements will be pursued through new optimizations in subsequent studies.

Fig. R7. (a) The force-displacement curves of the same sample during two cyclic loadings. (b) The wrinkling splitting occurring in two cyclic loadings. Scale bars: (b) $2\mu\text{m}$

Fig. R8. The solid balls indicate the normalized wrinkling wavelengths measured in the same sample during three cyclic loadings. The curves show the theoretical normalized wrinkling wavelengths versus shear strains with different bending stiffness.

These discussions have been added to Supplementary Materials, Pages 18-19, Lines 264-298.

Minor comments:

Q5. In Figure 2, it is inappropriate to show a mean and error on 4 graphene and 2 MoS2 devices, especially given the high spread. It would be better to just show a scatter plot of the individual values with the error for each device.

A5: Thanks. We have replotted Fig. 2f as scatter format (Manuscript, Page 22, Line 531).

Q6. Confusingly, the introduction, abstract and conclusion contain numerous grammatical issues, while the main body of the text is comparatively well written. Edit for grammar.

A6: We are sorry for these grammar issues. In the manuscript, we revised all the issues which are highlighted in revised version.

Response to comments of Reviewer #3:

Q1: The manuscript by Hou et al. reports experimental results for wrinkle formation in graphene and MoS₂ monolayer suspended over a trench, which is part of the device. Through a “push to shear” delicate mechanical process carried out under SEM, the authors observe formation of periodic wrinkles followed by the formation of a secondary wrinkle pattern of smaller wavelength. The mechanism of formation is wrinkle splitting, as observed also in the reported molecular dynamics simulations. Under release, graphene flattens directly without revisiting the primary wrinkle pattern of larger wavelength. While I believe that these results merit publication in some capacity, the manuscript, as a whole, does not possess the caliber required for acceptance in Nature Communications. Some of the reasons are listed below:

A1: We thank the referee for careful reading and thoughtful critique of the manuscript. We have carefully revised the manuscript to address your comments and concerns. Through this revision, we hope the novelty and significance of the study can be better highlighted in the current version.

Q2: The introductory part fails to be clearly articulate the importance of this work. Without a convincing explanation, the authors assert exaggerations without providing substantiating evidence: “Such binary instability, as a basic feature of compressed atomically thin films, is deemed to play a vital role in physical properties of 2D materials.” But authors do not elaborate why. The introduction is also written in a confusing manner and some paragraphs do not have a logical flow: “Besides, the static binary instability patterns, referring to the initial wrinkles could bifurcate into period-halving wrinkles, were experimentally discovered in few layers graphene and other membranes.” “We achieve in a shear-induced reversible binary instability (include primary and secondary instabilities) of monolayer 2D materials, which can be quantified by a series critical stresses that governed by pretension, geometry and modulus of materials.”

A2: Thank you for pointing out the shortcomings in the presentation of the importance of this work. We removed some inappropriate expressions, such as “Such binary instability, as a basic feature of compressed atomically thin films, is deemed to play a vital role in physical properties of 2D materials” And the sentences you mentioned above.

The introduction has been revised in Manuscript, Pages 3-4, Lines 50-80.

“Experimentally, apply controllable out-of-plane deformation for creating and tuning the instability in suspended 2D materials is rather challenging. Previous work utilized thermal stress

to achieve buckling compression on suspended graphene layers¹⁷, where the static periodic wrinkles and wrinklons (which referring to the initial wrinkles could split into period-halving wrinkles¹⁵) were observed in the buckled graphene. Similar phenomena were also discovered in other thin films^{16, 17, 18, 19}. However, the instability and recovery process of suspended monolayer 2D materials cannot yet be well quantified due to the absence of mechanical measurements. Besides, the transition between wrinkles and wrinklons, which plays a vital role in tuning instability behaviors of 2D materials^{20, 21, 22}, yet remains elusive in suspended monolayer 2D materials..... After unloading, does the recovery process backtrack the forward process? Quantitative experimental measurement of 2D materials instability is therefore urgently needed.”

“..... Compared to the realized micro/nano-scale mechanical experiments for 2D materials, e.g. tension, compression and bending experiments (Figs. 1a)^{17, 23, 24}, the in-plane shearing can precisely control the instability behavior of thin film..... In our measurement, we observed the rise of primary and secondary instabilities under increasing shear loadings. The loadings can be quantified by a series critical stresses that governed by pretension, geometry and modulus of materials.....While the recovery process involves a steady amplitude-decreasing-dominated smoothing rather than merging. Such distinct instability and recovery trajectories can be explained by the local compressive stress redistribution of monolayer 2D materials.....”

Q3: Additionally, some concepts are not properly used and this again makes the manuscript confusing and difficult to read: “Is the restabilization process symmetric with the instability process?” Probably what the authors want to say here is “After unloading, does the recovery process backtrack the forward process?” if so “restabilization” and “symmetric” are not the proper words. As the manuscript eventually clarifies, the answer is no. However, it is improper to call “symmetry breaking” the fact that the recovery process does not backtrack the forward process. Similarly, referring to wrinkle splitting later on as “bifurcation” is also confusing.

A3: Thanks to your clarification of the above important concepts, we have reworked the above statements in the revised manuscript. We removed those “unprecedented”, “restabilization” and “symmetry” words, and revised the manuscript to “different instability and recovery process”. In the introduction, we changed to “After unloading, does the recovery process backtrack the forward process?” (Manuscript, Page 3, Line 61). Besides, we used “*wrinkle splitting*” to describe observed experimental phenomenon in SEM images. The term “*bifurcation*” implies that “*wrinkle splitting*” marks the point of energy bifurcation, signifying the existence of different stress solutions under the same system energies. So now we used “*wrinkle splitting*” to describe our findings, which have been highlighted in the revised manuscript.

Q3: The values of the elastic constants extracted from this experiment are in very modest

agreement with what is expected from the previous literature. The shear modulus of 70 GPa is very low compared with what is reported in the experimental reference 29. The measured bending constants are large compared with theoretical predictions and other experiments. The provided explanation “this is expected given that thermal fluctuations and static ripples are expected to significantly stiffen micron-sized monolayer 2D” is not clear or convincing.

A4: Thank you for this important comment. Theoretically, if a monolayer 2D material (e.g., graphene) is considered as continuous film, its shear modulus can be estimated from Young’s modulus, $G = E/(2(1 + \nu))$, which is ~400 GPa (E is taken as 1000 GPa). Nevertheless, measurement of the shear modulus of thin films poses inherent challenges. This is primarily attributed to the fact that thin films are typically subjected to initial corrugations, leading to a diminished load-carrying capacity during the initial stages of shear loading. In our experiment, we calculate the shear modulus by analyzing the stress-strain relationship during the initial loading process. This process contains the transition from an initial undulated state to a flat state. The AFM data presented in the revised manuscript underscores that, the suspended 2D material exhibits corrugations ranging from a few nanometers to a dozen nanometers. Therefore, the measurements of shear modulus in our experiments may be lower than the theoretical values.

The discussion has been added in Manuscript, Page 6, Lines 122-124: “Besides, the low shear modulus measured in our experiment may be attributed to the fact that films are typically subjected to initial corrugations, which is discussed in *Supplementary Materials*, section 3.”

Conversely, the initial corrugations contribute to an augmentation in the bending stiffness. This phenomenon stands out as a primary factor explaining why our measured bending stiffness surpasses the theoretical value. Additionally, the revised manuscript delves into other factors, such as boundary slippage, irradiation damage, and measurement error etc.

These discussions have been added in Manuscript, Pages 9-10, Lines 194-204: “The results are higher than the value anticipated in atomic-scale measurement (~1 eV)³⁷. Here we consider the effect of measurement error and initial corrugations on bending stiffness measurements, and the detailed analysis can be seen in *Supplementary Materials*, section 3, Figs. S10-S13. We infer the major source of bending stiffness fluctuations is randomly distributed initial corrugations, which is similar to that thermal fluctuations and static ripples are expected to significantly stiffen micron-sized monolayer 2D materials⁹. According to theoretical calculation⁴⁰, the effective bending stiffness D_{eff} of the thin film with corrugations $\langle A^2 \rangle$ can be written as $D_{\text{eff}}/D_0 = k_B T W^2 / 16\pi D_0 \langle A^2 \rangle$, where $k_B T$ is the thermal energy, and D_0 is the intrinsic bending stiffness of 2D materials. Correspondingly, if the initial corrugations are 5 nm to 15 nm. the effective bending stiffness D_{eff} is 5 eV to 47 eV, which roughly agrees with the range of experimental measurement.”

These discussions have been added in *Supplementary Materials*, Pages 16-17, Lines 223-250.

Q4: I find it a stretch calling a “paradox” the fact that the recovery process does not follow the same path as the forward process. Of course, upon external decompression, the wrinkles should flatten directly – a paradox would be if they would instead localize into the larger amplitude and larger wavelength wrinkles. Also the forward cause of the secondary wrinkle is not different from the formation of the primary one, i.e. compression release into bending! This should not be presented as a surprise, as the current manuscript does.

A4: We think your comment on this point is very reasonable and we agree. Now, we have removed all the hyperbolic descriptions and changed the Manuscript as “different instability and recovery process”. All the modifications have been highlighted in revised Manuscript.

Q5: The scaling law derived from the continuum analysis seems to have some similarity with the continuum analysis of Cerda and Mahadevan in Phys. Rev. Lett. 90, 074302. perhaps the authors could have a look at this paper to make better contact with previous understanding.

A5: Thank you for pointing out this important literature. We carefully read and compared the literature and the models in this article. The literature shows a very clear and concise power law relationship between wavelength and stiffness. In the analysis of Cerda and Mahadevan, wrinkles occur in uniaxial stretched rectangular elastic membranes to accommodate the in-plane strain incompatibility caused by the Poisson effect, wherein the transverse compression induced during longitudinal stretching serves as the origin of wrinkling instability. Given by the high in-plane stiffness of 2D materials, it is difficult to observe wrinkling instability induced by uniaxial stretching. In our model, wrinkles originate from the compressive stress during the push-to-shearing process, and the prestrain was considered here to enhance the correlation with experimentally measured parameters and theoretical model. Although the wrinkling mechanism is different, the bending and stretching energy in both cases follow the same scaling with respect to the wavelength, i.e., $U_b \sim D_0 A^2 W / \lambda^3$, and $U_s \sim Et \varepsilon A^2 \lambda / W$, where A is the amplitude of wrinkle, W is the membrane size, and ε is the in-plane strain. By minimizing the total energy with respect to the wavelength, one can obtain the scaling law between bending stiffness and wavelength length, $\lambda \sim (D_0 / Et \varepsilon)^{1/4}$. This reference has been cited in Manuscript as Ref. 34. With this, we hope our revised manuscript can stimulate more research interests on this potentially very important topic.

REVIEWERS' COMMENTS

Reviewer #1 (Remarks to the Author):

The authors addressed the reviewer's comments in an exhaustive manner; the manuscript can now be accepted for publication.

Reviewer #2 (Remarks to the Author):

I applaud the authors on all the new measurements and analysis performed in the revision, which significantly enhance the depth of understanding in the work. I particularly appreciate the correlation between fluctuations and analysis of the bending stiffness. There are a few remaining comments which need to be addressed.

1. While the introduction is better written in general, the authors did not address one of my main comments. As discussed in my previous review, shear induced wrinkling in graphene has been observed and analyzed previously, but this continues to be obfuscated. The authors need to explicitly include a few sentences reviewing work on in-plane shear *and what was learned from those works* to give an accurate representation of the field and what is actually new in their work. (e.g. Ref 17, which is not just compression as stated in the manuscript).

2. In the new supporting figure S12, only the second and third loadings are shown, but all three loadings are discussed. This is a very odd choice. The authors need to show all three loadings to give confidence in the interpretation.

Reviewer #3 (Remarks to the Author):

I find that the author responses are reasonable and I recommend publication of the recomposed manuscript.

Responses to Reviewers' Comments (NCOMMS-23-42466A)

Reviewer #1: The authors addressed the reviewer's comments in an exhaustive manner; the manuscript can now be accepted for publication.

Reply: We are grateful to the reviewer for the review and support.

Reviewer #2: I applaud the authors on all the new measurements and analysis performed in the revision, which significantly enhance the depth of understanding in the work. I particularly appreciate the correlation between fluctuations and analysis of the bending stiffness. There are a few remaining comments which need to be addressed.

Reply: We appreciate the reviewer for the positive feedback and detailed review, which have greatly contributed to the improvement of our manuscript.

*A1. While the introduction is better written in general, the authors did not address one of my main comments. As discussed in my previous review, shear induced wrinkling in graphene has been observed and analyzed previously, but this continues to be obfuscated. The authors need to explicitly include a few sentences reviewing work on in-plane shear *and what was learned from those works* to give an accurate representation of the field and what is actually new in their work. (e.g. Ref 17, which is not just compression as stated in the manuscript).*

Reply: Thank you for the suggestion. We do appreciate those earlier related works and tried to highlight the novelty of our present study. According to your valuable suggestion, we now have changed the relevant part of our Introduction as “Previous work firstly achieved controllable buckling of suspended graphene layers by controlling boundary conditions and utilizing graphene’s negative thermal expansion coefficient¹⁷. They further observed both of compression- and shearing-induced static periodic wrinkles and wrinklons (which referring to the initial wrinkles could split into period-halving wrinkles¹⁸) in the buckled graphene, which is induced by spontaneously and/or thermally generated strains¹⁷.” in Manuscript Page 3 Lines 52-57. With that, we hope our new approach can also contribute to this important field.

A2. In the new supporting figure S12, only the second and third loadings are shown, but all three loadings are discussed. This is a very odd choice. The authors need to show all three loadings to give confidence in the interpretation.

Reply: Thanks. The 1st cyclic load-unloading curve has been added in Supplementary Fig. 13.

Reviewer #3: I find that the author responses are reasonable and I recommend publication of the recomposed manuscript.

Reply: Thank you very much for your positive feedback and the recommendation for publication. We are grateful to you for giving all the valuable suggestions and comments for this work.